# The Absence of C-5 DNA Methylation in *Leishmania donovani* Allows DNA Enrichment from Complex Samples

**DOI:** 10.3390/microorganisms8081252

**Published:** 2020-08-18

**Authors:** Bart Cuypers, Franck Dumetz, Pieter Meysman, Kris Laukens, Géraldine De Muylder, Jean-Claude Dujardin, Malgorzata Anna Domagalska

**Affiliations:** 1Molecular Parasitology, Institute of Tropical Medicine, 2000 Antwerp, Belgium; bart.cuypers@uantwerpen.be (B.C.); fd353@cam.ac.uk (F.D.); geraldine_demuylder@hotmail.com (G.D.M.); jcdujardin@itg.be (J.-C.D.); 2ADReM Data Lab, Department of Computer Science, University of Antwerp, 2000 Antwerp, Belgium; pieter.meysman@uantwerpen.be (P.M.); kris.laukens@uantwerpen.be (K.L.); 3Department of Biomedical Sciences, University of Antwerp, 2000 Antwerp, Belgium

**Keywords:** *Leishmania*, trypanosomatids, DNA-methylation, epigenomics, whole-genome bisulfite sequencing, DNA-enrichment

## Abstract

Cytosine C5 methylation is an important epigenetic control mechanism in a wide array of eukaryotic organisms and generally carried out by proteins of the C-5 DNA methyltransferase family (DNMTs). In several protozoans, the status of this mechanism remains elusive, such as in *Leishmania*, the causative agent of the disease leishmaniasis in humans and a wide array of vertebrate animals. In this work, we showed that the *Leishmania donovani* genome contains a C-5 DNA methyltransferase (*DNMT*) from the *DNMT6* subfamily, whose function is still unclear, and verified its expression at the RNA level. We created viable overexpressor and knock-out lines of this enzyme and characterized their genome-wide methylation patterns using whole-genome bisulfite sequencing, together with promastigote and amastigote control lines. Interestingly, despite the DNMT6 presence, we found that methylation levels were equal to or lower than 0.0003% at CpG sites, 0.0005% at CHG sites, and 0.0126% at CHH sites at the genomic scale. As none of the methylated sites were retained after manual verification, we conclude that there is no evidence for DNA methylation in this species. We demonstrated that this difference in DNA methylation between the parasite (no detectable DNA methylation) and the vertebrate host (DNA methylation) allowed enrichment of parasite vs. host DNA using methyl-CpG-binding domain columns, readily available in commercial kits. As such, we depleted methylated DNA from mixes of *Leishmania* promastigote and amastigote DNA with human DNA, resulting in average *Leishmania:*human enrichments from 62× up to 263×. These results open a promising avenue for unmethylated DNA enrichment as a pre-enrichment step before sequencing *Leishmania* clinical samples.

## 1. Introduction

DNA methylation is an epigenetic mechanism responsible for a diverse set of functions across the three domains of life; eubacteria, archaebacteria, and eukaryota. In prokaryotes, many DNA methylation enzymes are part of so-called restriction-modification systems, which play a crucial role in their defense against phages and viruses. Prokaryotic methylation typically occurs on the C5 position of cytosine (cytosine C5 methylation), the exocyclic amino groups of adenine (adenine-N6 methylation), or cytosine (cytosine-N4 methylation) [1]. In eukaryotic species, DNA methylation is mostly restricted to 5-methylcytosine (me^5^C) and best characterized in mammals, where 70–80% of the CpG motifs are methylated [2]. As such, DNA methylation controls a wide range of important cellular functions, such as genomic imprinting, X-chromosome inactivation (in humans), gene expression, and the repression of transposable elements. Consequently, defects in genetic imprinting are associated with a variety of human diseases, and changes in DNA methylation patterns are a common hallmark of cancer [3,4]. Eukaryotic DNA methylation can also occur at CHG and CHH (where H is A, C, or T) sites [5], which was considered to occur primarily in plants. However, studies from the past decade demonstrate that CHG and CHH methylations are also frequent in several mammalian cell types, such as embryonic stem cells, oocytes, and brain cells [5,6,7,8]. 

Me^5^C methylation is mediated by a group of enzymes called C-5 DNA methyltransferases (DNMTs). This ancient group of enzymes shares a common ancestry, and their core domains are conserved across prokaryotes and eukaryotes [1]. Different DNMT subfamilies have developed distinct roles within epigenetic control mechanisms. For example, in mammals, DNMT3a and DNMT3b are responsible for de novo methylation, such as during germ cell differentiation and early development, or in specific tissues undergoing dynamic methylation [9]. In contrast, DNMT1 is responsible for maintaining methylation patterns, particularly during the S phase of the cell cycle, where it methylates the newly generated hemimethylated sites on the DNA daughter strands [10]. Some DNMTs have also changed substrate over the course of evolution. A large family of DNMTs, called DNMT2, has been shown to methylate the 38th position of different tRNAs to yield ribo-5-methylcytidine (rm5C) in a range of eukaryotic organisms, including humans [11], mice [12], *Arabidopsis thaliana* [13], and *Drosophila melanogaster* [14]. Therefore, DNMT2s are now often referred to as ‘tRNA methyltransferases’ or trDNMT and are known to carry out diverse regulatory functions [15]. However, in other eukaryotic taxa, DNMT2 appears to be a genuine DNMT, as DNMT2 can catalyze DNA methylation in *Plasmodium falciparum* [16] and *Schistosoma mansoni* [17]. In *Entamoeba histolytica,* both DNA and RNA can be used as substrates for DNMT2 [18,19]. The increase in available reference genomes of non-model eukaryotic species has recently resulted in the discovery of new DNMTs, such as DNMT5, DNMT6, or even SymbioLINE-DNMT—a massive family of DNMTs, so far only found in the dinoflagellate *Symbiodinium* [20]. 

Indeed, DNMT-mediated C5 methylation has been shown to be of major functional importance in a wide array of eukaryotic species, including protozoans, such as *Toxoplasma gondii* and *Plasmodium* [16,21]. In contrast, studies have failed to detect any C5 DNA methylation in eukaryotic species, such as *Caenorhabditis elegans, Saccharomyces cerevisiae,* and *Schizosaccharomyces pombe* [22,23]. In many other protozoans, the presence and potential role of DNA methylation remain elusive. This is especially true for *Leishmania*, a trypanosomatid parasite (Phylum Euglenozoa), despite its medical and veterinary importance. *Leishmania* is the causative agent of leishmaniasis—a disease that ranges from self-healing cutaneous lesions to lethal visceral leishmaniasis—in humans and a wide variety of vertebrate animals. 

*Leishmania* features molecular biology that is remarkably different from other eukaryotes. This includes a system of polycistronic transcription of functionally unrelated genes [24]. The successful transcription of these cistrons depends on several known epigenetic modifications at the transcription start sites (acetylated histone H3) and transcription termination sites (β-D-glucosyl-hydroxymethyluracil, also called ‘Base J’). However, little research has been done towards other epigenetic modifications [25]. We were, therefore, interested in the 5-C methylation status of *Leishmania,* which has been poorly explored to date. In this context, a single study on a wide range of eukaryotic species lacking DNMT1 has reported the absence of CG-specific methylation in *Leishmania major*, however, using only a single sample of an unspecified life stage [26]. The study also does not comment on CHH and CHG specific methylation, which can be relevant as well. Contrastingly, Militello et al. demonstrated Me^5^C methylation in *Trypanosoma brucei,* another trypanosomatid species, although at low levels (0.01%) [27]. To clarify the status of C-5 DNA methylation in trypanosomatids and *Leishmania,* in particular, we presented the first comprehensive study of genomic methylation in *Leishmania* across different parasite life stages, making use of high-resolution whole-genome bisulfite sequencing. 

## 2. Materials and Methods

### 2.1. In Silico Identification and Phylogeny of Putative DNMTs

To identify putative C-5 cytosine-specific DNA methylases in *Leishmania donovani*, we obtained the hidden Markov model (hmm) for this protein family from PFAM version 32.0 (Accession number: PF00145) [28]. The hmm search tool of hmmer-3.2.1 (hmmer.org) was then used with default settings to screen the LdBPKV2 reference genome [29] for this hmm signature. The initial pairwise alignment between the identified *L. donovani* and *T. brucei* C5 DNA MTase was carried out with T-COFFEE V_11.00.d625267.

To construct a comprehensive phylogenetic tree of the C5 DNA MTase family, including members found in trypanosomatid species, we modified the approach from Ponts et al. [16]. Firstly, we downloaded the putative proteomes of a wide range of prokaryotic and eukaryotic species. These species were selected to cover the different C5 DNA MTase subfamilies [1]. Specifically, the following proteomes were obtained: *Leishmania amazonensis* MHOMBR71973M2269 [30], *Leishmania braziliensis* MHOMBR75M2904 [31], *Leishmania major* Friedlin [24], *Leishmania tarentolae* Parrot-TarII [32], *Leptomonas seymouri* ATCC30220 [33], *Trypanosoma brucei* TREU92 [34], *Trypanosoma congolense* IL3000_2019 [35], *Trypanosoma cruzi* CL Brener [20], *Trypanosoma grayi* ANR4 [36] and *Trypanosoma vivax* Y486 from TriTrypDB v46 [37], *Plasmodium falciparum* 3D7 and *Plasmodium vivax* P01 from PlasmodDB v46 [38,39,40], *Cryptosporidium parvum* Iowa II and *Cryptosporidium hominis* TU502 from CryptoDB v46 [41,42,43], *Toxoplasma gondii* ARI from ToxoDB v 46 [44,45], *Agrobacterium tumefaciens* (GCF_000971565.1) [46], *Arabidopsis thaliana* (GCF_000001735.4), *Aspergillus fumigatus* (GCF_000002655.1), *Aspergillus nidulans* FGSC A4 (GCF_000149205.2) [47], *Bacillus subtilis* 168 (GCF_000009045.1), *Clostridium botulinum* ATCC 3502 (GCF_000063585.1) [48], *Drosophila melanogaster* (GCF_000001215.4), *Emiliana huxleyi* (GCF_000372725.1) [49], *Entamoeba histolytica* HM-1:IMSS (GCF_000208925.1) [50], *Escherichia coli* K12 (GCF_000005845.2), *Eubacterium rectale* (GCF_000020605.1), *Euglena gracilis* Z1 (PRJNA298469) [51], *Micromonas pusilla* (GCF_000151265.2) [52], *Neurospora crassa* OR74A (GCF_000182925.2) [53], *Nostoc punctiforme* PCC 73102 (GCF_000020025.1), *Homo sapiens* GRCh38.p12 (GCF_000001405.38), *Phaeodactylum tricornutum* (GCF_000150955.2) [54], *Saccharomyces cerevisiae* S288C (GCF_000146045.2), *Salmonella enterica* CT18 (GCF_000195995.1) [55], *Schizosaccharomyces pombe* ASM294 (GCF_000002945.1) [56], *Streptococcus pneumoniae* R6 (GCF_000007045.1) [57], *Thalassiosira pseudonana* (GCF_000149405.2) and *Treponema succinifaciens* (GCF_000195275.1) [58] from NCBI, *Ascobolus immersus* RN42 [59] from the JGI Genome Portal (genome.jgi.doe.gov), and *Danio rerio* (GRCz11) from Ensembl (ensembl.org). 

All obtained proteomes were then searched with the hmm signature for C5 DNA MTases, exactly as described above for *L. donovani.* All hits with an E-value <0.01 (i.e., 1 false-positive hit is expected in every 100 searches with different query sequences) were maintained, and all domains matching the query hmm were extracted and merged per protein. This set of sequences was aligned in Mega-X with the MUSCLE multiple sequence alignment algorithm [60,61] and converted to the PHYLIP format with the ALTER tool [62]. Phage sequences and closely related isoforms were removed for the phylogenetic tree to maintain visibility.

A maximum-likelihood tree of this alignment was generated with RAxML version 8.2.10 using the automatic protein model assignment algorithm (option: -m PROTGAMMAAUTO). RAxML was run in three steps: Firstly, 20 trees were generated, and only the one with the highest likelihood score was kept. Secondly, 1000 bootstrap replicates were generated. In a final step, the bootstrap bipartitions were drawn on the best tree from the first round. The tree was visualized in Figtree v1.4.4 (https://github.com/rambaut/figtree/). 

### 2.2. Culturing and DNA Extraction for Bisulfite Sequencing 

Promastigotes (extracellular life stage) of *Leishmania donovani* MHOM/NP/03/BPK282/0 cl4 (further called BPK282) and its genetically modified daughter lines (see below) were cultured in HOMEM (Gibco, Thermo Fisher Scientific, Waltham, MA, USA) supplemented with 20% (*v*/*v*) heat-inactivated fetal bovine serum at 26 °C. Amastigotes (intracellular life stage) of the same strain were obtained from three months’ infected golden Syrian hamster (Charles Rivers, Wilmington, MA, USA), as described in Dumetz et al. [29], based on Pescher et al. [63] and respecting BM2013-8 ethical clearance from Institute of Tropical Medicine (ITM) Animal Ethics Committee. Briefly, 5-week-old female golden hamsters were infected via intracardiac injection of 5 × 10^5^ stationary phase promastigotes. Three months post-infection, hamsters were euthanized, and amastigotes were purified from the liver. The liver was homogenized in HOMEM, then cleared by centrifugation (130× *g* for 5 at room temperature). The supernatant was collected, and saponin lysed (1 mg × mL^−1^) under agitation for 5 min, then centrifuged for 10 min at 1800× *g*. The pellet was washed three times with ice-cold PBS. Remaining host cells were removed by Percoll centrifugation (GE Healthcare, Chicago, IL, USA) in a 15 mL Falcon tube: the parasites were resuspended in a 45% Percoll solution layered on a 90% Percoll cushion. The gradient interface was recovered after 30 min centrifugation at 3500× *g*, 15 °C, then washed 3 times with ice-cold PBS (1800× *g*, 10 min, 15 °C) prior to DNA extraction. *T. brucei gambiense* MBA bloodstream forms were obtained from OF-1 mice when the parasitemia was at its highest, according to ITM Animal Ethics Committee decision BM2013-7 (26/02/2013). Parasites were separated from the whole blood, as described in Tihon et al. [64]. Briefly, the parasites were separated from the blood by placing the whole blood on an anion exchanger diethylaminoethyl (DEAE)-cellulose resin (Whatman, Maidstone, UK) suspended in phosphate saline glucose (PSG) buffer, pH 8. After elution and two washes on PSG, DNA was extracted. DNA of *L. donovani*, both promastigotes and amastigotes, as well as *T. brucei,* was extracted using DNeasy Blood and Tissue Kit (Qiagen, Hilden, Germany) according to manufacturer instructions. 

*Arabidopsis thaliana* Col-0 was grown for 21 days under long-day conditions, i.e., 16 h light and 8 h darkness. DNA was then extracted from the whole rosette leaves using the DNeasy Plant Mini Kit (Qiagen).

### 2.3. Genetic Engineering of L. donovani BPK282

We generated both an LdDNMT overexpressing (LdDNMT+) and the null mutant line (LdDNMT-/-) of *L. donovani* BPK282. All the PCR products generated to produce the constructs for LdDNMToverex and LdDNMTKO were sequenced at the VIB sequencing facility using the same primer as for the amplification. For LdDNMToverex, the overexpression construct, pLEXSY-DNMT, was generated by PCR amplification of LdBPK_251230 from BPK282 genomic DNA using Phusion (NEB, Ipswich, Massachusetts, US) and cloned inside the expression vector pLEXSY-Hyg2 (JENA Bioscience, Jena, Germany) using NEBuilder (NEB) according to manufacturer’s instruction for primer design and cloning instructions (Appendix A for primers list). Once generated, 10 µg of pLEXSY-DNMT was electroporated in 5 × 10^7^ BPK282 promastigotes from logarithmic culture using cytomix on a GenePulserX (BioRad, Hercules, California, US), according to LeBowitz (1994) [65], and selected in vitro by adding 50 μg/mL hygromycin B (JENA Bioscience) until parasite growth [66]. Verification of overexpression was carried out by qPCR on a LightCycler480 (Roche, Basel, Switzerland) using SensiMix SYBR No-ROX (Bioline, London, UK) on cDNA. Briefly, 10^8^ logarithmic-phase promastigotes were pelleted, RNA extraction was performed using RNAqueous-Micro Total RNA Isolation Kit (Thermo Fisher Scientific) and quantified by Qubit, and the Qubit RNA BR Assay (Thermo Fisher ScientificTranscriptor reverse transcriptase (Roche) was used to synthesize cDNA following manufacturer’s instructions. qPCRs were run on a LightCycler 480 (Roche) with a SensiMix SYBR No-ROX Kit (Bioline); primer sequences are available in Appendix A. Normalization was performed using two transcripts, previously described as stable in promastigotes and amastigotes by Dumetz et al. (2018) [67]—LdBPK_340035000 and LdBPK_240021200. 

For the generation of LdDNMT-/-, a two-step gene replacement strategy was used: replacing the first allele of LdBPK_251230 by nourseothricin resistance gene (SAT) and the second allele by a puromycin resistance gene (Puro). Briefly, each drug resistance gene was PCR amplified from pCL3S and pCL3P using Phusion (NEB) and cloned between 300 bp of PCR amplified DNA fragments from the upstream and downstream region of LdBPK_251230 using NEBuilder (NEB) inside pUC19 for construct amplification in *E. coli* DH5α (Promega, Madison, WI, USA) (cf. primer list in Appendix A). Each replacement construct was excised from pUC19 using SmaI (NEB), dephosphorylated using Antarctic phosphatase (NEB), and 10 µg of DNA was used for the electroporation in the same conditions, as previously described, to insert the pLEXSY-DNMT. The knock-out was confirmed by whole-genome sequencing. 

### 2.4. Bisulfite Sequencing and Data Analysis

For each sample, one microgram of genomic DNA was used for bisulfite conversion with innuCONVERT Bisulfite All-In-One Kit (Analytikjena, Jena, Germany). Sequencing libraries were prepared with the TruSeq DNA Methylation Kit according to the manufacturer’s instructions (Illumina, San Diego, CA, USA). The resulting libraries were paired-end (2 × 100 bp) and sequenced on the Illumina HiSeq 1500 platform of the University of Antwerp (Centre of Medical Genetics). The sequencing quality was first verified with FastQC v0.11.4. Raw reads generated for each sample were aligned to their respective reference genome with BSseeker 2-2.0.3 [68]: LdBPK282v2 [29] for *L. donovani*, TREU927 [34] for *T. brucei,* and Tair10 [69] for the *A. thaliana* positive control. Samtools fixmate (option -m) and samtools markdup (option -r) were then used to remove duplicate reads. CpG, CHG, and CHH methylation sites were subsequently called with the BS-Seeker2 ‘call’ tool using default settings and further filtered with our Python3 workflow called ‘Bisulfilter’ (available at https://github.com/CuypersBart/Bisulfilter). Genome-wide visualization of methylated regions was then carried out with ggplot2 in R [70]. In *Leishmania*, the positions that passed our detection thresholds (coverage >25, methylation percentage >0.8) were then manually inspected in IGV 2.5.0 [71]. 

### 2.5. Leishmania DNA Enrichment from A Mix of Human and Leishmania DNA

To check whether the lack of detectable DNA methylation in *Leishmania* can be used for the enrichment of *Leishmania* vs. (methylated) human DNA, we carried out methylated DNA removal on two types of samples: (1) An artificial mix of *L. donovani* BPK282/0 cl4 promastigote DNA with human DNA (Promega) from 1/15 to 1/150,000 (*Leishmania*:human) and (2) Linked promastigote and hamster-derived amastigote samples from 3 clinical *Leishmania donovani* strains (BPK275, BPK282, and BPK026), which were generated in previous work [29]. For this experiment, we used a 1/1500 artificial mix of promastigote DNA and human DNA (Promega) to reflect the median ratio found in clinical samples. For each of the three biological replicates (strains), we carried out the experiment in duplicate (technical replicates). All parasite DNA was extracted with the DNA (DNeasy Blood and Tissue Kit, Qiagen). *Leishmania* DNA (0.017 ng/μL) was then enriched from the human DNA (25 ng/μL) using NEBNext Microbiome DNA Enrichment Kit (NEB) according to manufacturer instructions. Evaluation of the ratio *Leishmania*/human DNA was performed by qPCR on LightCycler480 (Roche) using SensiMix SYBR No-ROX (Bioline) and RPL30 primers provided in the kit to measure human DNA and *Leishmania* CS primers (cysteine synthase) [72]. Cysteine synthase is a single copy gene localized on chromosome 36. Across all the different studies carried on *L. donovani,* chromosome 36 is shown to be disomic in all strains (Imamura et al. 2016). *Leishmania* DNA quantification using qPCR and targeting the CS gene was previously validated by sequencing by Domagalska et al. [73].

## 3. Results

### 3.1. The Leishmania Genome Contains a Putative C-5 DNA Methyltransferase (DNMT)

Eukaryotic DNA methylation typically requires the presence of a functional C-5 cytosine-specific DNA methylase (C5 DNA MTase). This type of enzymes specifically methylates the C-5 position of cytosines in DNA, using S-adenosyl methionine as a methyl-donor. To check for the presence of C5 DNA MTases in *Leishmania donovani*, we carried out a deep search of the parasite’s genome. In particular, we used the LdBPKv2 reference genome [29] and searched the predicted protein sequences of this assembly using the hidden Markov model (hmm) signature of the C5 DNA MTase protein family obtained from PFAM (PF00145) and obtained a single hit: the protein LdBPK_251230 (E-value: 2.7 × 10^−40^). LdBPK_251230 was already annotated as ‘modification methylase-like protein’ with a predicted length of 840 amino acids. In the following text, this protein has been referred to as LdDNMT. Interestingly, in another trypanosomatid species, *Trypanosoma brucei*, the homolog of this protein (Tb927.3.1360 or TbDNMT) has been previously studied in detail by Militello et al. [27]. Moreover, these authors showed that TbDNMT has all ten conserved domains that are present in functional DNMTs. We aligned TbDNMT with LdDNMT using T-Coffee (Figure 1) and found that these 10 domains are also present in LdDNMT, including the putative catalytic cysteine residue in domain IV. 

### 3.2. Leishmania and Trypanosomatid C-5 DNA Methyltransferase Belongs to the Eukaryotic DNMT6 Family

To learn more about the putative function and evolutionary history of this protein, we wanted to characterize the position of *LdDNMT* and those of related trypanosomatid species within the DNMT phylogenetic tree. Consequently, we collected the publicly available, putative proteomes of a wide range of prokaryotic and eukaryotic species, searched them for the hmm signature of the C5-DNMT family, aligned the identified proteins, and generated a RAxML maximum likelihood tree. In total, we identified 211 putative family members in the genomes of 37 species (E-value < 0.01), including seven prokaryotic (*Agrobacterium tumefaciens, Clostridium botulinum, Escherichia coli*, *Eubacterium rectale, Nostoc punctiforme, Salmonella enterica,* and *Treponema succinifaciens*) and 30 eukaryotic species. These eukaryotic species were selected to contain organisms from the excavata phylum (of which *Leishmania* is part) and a range of other, often better-characterized phyla as a reference. The excavata species included 11 trypanosomatids (*Trypanosoma grayi, Trypanosoma congolense, Trypanosoma vivax, Trypanosoma brucei, Trypanosoma cruzi, Leptomonas seymouri, Leishmania braziliensis, Leishmania tarentolae, Leishmania amazonensis, Leishmania major, and Leishmania donovani*), 1 other non-trypanosomatid euglenozoid species (*Euglena gracilis*), and 1 other non-euglenozoid species (*Naegleria gruberi*). The other eukaryotic phyla included in the analysis were apicomplexa (*Plasmodium vivax*, *Plasmodium falciparum*, *Cryptosporidium parvum*, *Cryptosporidium hominis*), amoebozoa (*Entamoeba histolytica*), angiosperma (*Arabidopsis thaliana*, *Oryza sativa*), ascomycota (*Ascobolus immerses*, *Aspergillus fumigatus*, *Aspergillus nidulans, Neurospora crassa*), chlorophyta (*Micromonas pusilla*), chordata (*Homo Sapiens, Danio rerio),* haptophyta (*Emiliania huxleyi*), ochrophyta (*Phaeodactylum tricornutum*, *Thalassiosira pseudonana*) (Figure 2). 

Our phylogenetic tree was able to clearly separate known DNMT subgroups, including DNMT1, DNMT2, DNMT3, DNMT4, DNMT5, and several groups of prokaryotic DNMTs [1,16,74]. Interestingly, the tree also showed that most trypanosomatids have exactly one DNMT, which is part of the much less-characterized DNMT6 group, as has been previously described for *Leishmania major* and *Trypanosoma brucei* [20]. This group of DNMTs has also been found in chlorophyta, haptophyta, ochrophyta (in our tree represented as *Micromonas pusilla*, *Emiliania huxleyi,* and *Phaeodactylum tricornutum,* respectively) and recently in dinoflagellates (e.g., *Symbiodinium kawagutii* and *Symbiodinium minutum*), but its function remains elusive [20,75]. Next to this array of unicellular eukaryotes, the most closely related branch to DNMT6 contains a series of bacterial DNMTs (in our tree represented by *Agrobacterium tumefaciens, Eubacterium rectale, Escherichia coli, Salmonella enterica,* and *Treponema succinifaciens*). This suggests that DNMT6 already diversified from the other DNMT groups within the pool of prokaryotic DNMTs. Interestingly, from our tree, the DNMT composition of other excavata species appeared to be very different. The euglenozoid, *Euglena gracilis,* has DNMT1, DNMT2, DNMT4, and DNMT5, while *Naegleria gruberi* has both a DNMT1 and a DNMT2. In contrast, we found that the majority of trypanosomatids have just one single DNMT6. This complex distribution of DNMT groups over the different excavata species is suggestive of extensive horizontal gene transfer, which is regarded upon as one of the main mechanisms for the acquisition of new MTases in eukaryotes [20]. 

Interestingly, *Trypanosoma cruzi* was the only trypanosomatid with more than one DNMT, having two proteins significantly match the hmm. These two DNMTs are not part of the DNMT6 cluster, but instead had no close relatives in the tree (the closest being prokaryotic DNMTs). Importantly, the protein database used to identify these two DNMTs originated from a six-frame translation of the *T. cruzi* Bug2148 draft assembly. Therefore, further improvement of this draft proteome is needed to confirm or reject our findings. 

### 3.3. Whole-Genome Bisulfite Sequencing Reveals No Evidence for Functional C-5 Methylation

As (1) we identified LdBPK_251230 to be from the C5 DNA MTase family, (2) all 10 conserved domains were present. We decided to check for the presence and functional role of C5 DNA methylation in *L. donovani*. Therefore, we assessed the locations and degree of CpG, CHG, and CHH methylation across the entire *Leishmania* genome and within the two parasite life stages: amastigotes (intracellular mammalian life stage) and promastigotes (extracellular, insect life stage). Amastigotes were derived directly from infected hamsters, while promastigotes were obtained from axenic cultures. Promastigotes were divided into two batches, one passaged long-term in axenic culture, the other passaged once through a hamster and then sequenced at axenic passage 3; thus, allowing us to study also the effect of long vs. short term in vitro passaging. *Arabidopsis thaliana* and *T. brucei* were included as a positive control as the degree of CpG, CHG, and CHH methylation in *A. thaliana* is well known [76,77], while *T. brucei* is the only trypanosomatid in which (low) methylation levels were previously detected by mass spectrometry [27].

An overview of all sequenced samples can be found in Appendix A. All *L. donovani* samples were sequenced with at least 30 million 100 bp paired-end (PE) reads (60 million total) per sample, resulting in an average genomic coverage of at least 94X for the *Leishmania* samples. The *T. brucei* was sequenced with 69 million PE reads, resulting in 171X average coverage, and *A. thaliana* 27 million PE reads, resulting in 21X average coverage. Detailed mapping statistics can be found in Appendix A.

We first checked for global methylation patterns across the genome. Interestingly, we could not detect any methylated regions in *Leishmania donovani* promastigotes, both short (P3) and long-term in vitro passaged, nor in hamster-derived promastigotes or amastigotes (Figure 3). Minor increases in the CHH signal towards the start end of several chromosomes were manually checked in IGV and attributed to poor mapping in (low complexity) telomeric regions. This was in contrast to our positive control, *Arabidopsis thaliana*, which showed clear, highly methylated CpG, CHG, and CHH patterns across the genome. This distribution was consistent with prior results with MethGO observed on *Arabidopsis thaliana*, confirming that our methylation detection workflow was working [70].

In a second phase, we checked for individual sites that were fully methylated (>80% of the sequenced DNA at that site) using BS-Seeker2 and filtering the results with our automated Python3 workflow. CpG methylation in all three biological samples for *L. donovani* was lower than 0.0003%, CHG methylation lower than 0.0005%, and CHH methylation lower than 0.0126% (Table 1, Appendix A). However, when this low number of detected ‘methylated’ sites was manually verified in IGV, they could all clearly be attributed to regions where BS-Seeker2 wrongly called methylated bases, either because of poor mapping (often in repetitive, low complexity regions) or of strand biases. In reliably mapped regions, there was clearly no methylation. Similarly, we detected 0.0001% of CpG methylation, 0.0006% of CHG methylation, and 0.0040% of CHH methylation for *T. brucei*, which could all be attributed to mapping errors or strand biases. In *A. thaliana*, our positive control, we detected 21.05% of CpG methylation, 4.04% of CHG methylation, and 0.31% of CHH methylation, which is similar as the reported values in the literature [78,79], and demonstrated that our bioinformatic workflow could accurately detect methylated sites. We also checked sites with a lower methylation degree (>40%), which gave higher percentages, but this could be attributed to the increased noise level at this resolution (Appendix A). Indeed, even when applying stringent coverage criteria (>25×), this approach is susceptible to false-positive methylation calls, as we are checking millions of positions (in case of *Leishmania*, more than 5.8 million CG sites, 3.9 million CHG sites, and 9.3 million CHH sites).

To determine whether *LdDNMT* is essential, we generated an LdDNMT knock-out line (LdDNMT-/-) and validated it by calculating its *LdDNMT* copy number based on the sequencing coverage (Figure 4). Indeed, the copy number of the *LdDNMT* gene in LdDNMT-/- was reduced to zero. The fact that the LdDNMT-/- line was viable shows that *LdDNMT* is not an essential gene in promastigotes. Similar to the wild-type promastigotes and amastigotes described above, no CpG, CHG, or CHH methylation could be observed with bisulfite sequencing (Table 1, Appendix A). Vice versa, to check if overexpressing LdDNMT would affect the C-5 DNA-methylation patterns, we generated an overexpressor line (LdDNMT+). In this line, the LdDNMT genomic copy number was increased to 78 copies, and mRNA levels (Table 2) showed a 2.5-fold higher expression than the corresponding wild type. Although the LdDNMT+ initially seemed to have slightly higher methylation percentages (Table 1, Appendix A), none of the CpG, CHG, or CHH showed to be methylated after our manual validation in IGV, just as in the wild type lines and LdDNMT-/-. This suggested again that LdDNMT had no impact on DNA methylation, although we did not validate LdDNMT overexpression in LdDNMT+ on the protein level. Finally, neither the deletion nor the overexpression of LdDNMT had an obvious effect on the growth of the parasites, although this was not tested in a systematic way.

### 3.4. Absence of C5 DNA Methylation as a Leishmania vs. Host DNA Enrichment Strategy

The lack or low level of C5 DNA methylation opens the perspective for enriching *Leishmania* DNA in mixed parasite-host DNA samples, based on the difference in methylation status (the vertebrate host does show C5 DNA methylation). This could potentially be an interesting pre-enrichment step before the whole-genome sequencing analysis of clinical samples containing *Leishmania*. Furthermore, commercial kits for removing methylated DNA are readily available and typically contain a methyl-CpG-binding domain (MBD) column, which binds methylated DNA while allowing unmethylated DNA to flow through.

To test if these kits can be used for the relative enrichment of *Leishmania* DNA, we first generated artificially mixed samples using different ratios of *L. donovani* promastigote DNA with human DNA. Ratios were made starting from 1/15 to 1/15,000, which reflects the real ratio of *Leishmania* vs. human DNA in clinical samples [80]. From these mixes, *Leishmania* DNA was enriched using NEBNext Microbiome DNA Enrichment Kit (NEB) that specifically binds methylated DNA, while the non-methylated remains in the supernatant. *Leishmania* vs. human DNA ratios were determined by qPCR, targeting the RPL30 primers provided in the kit to measure human DNA and cysteine synthase for *Leishmania*. We observed an average of 263× enrichment of *Leishmania* vs. human DNA (Figure 5). This ranged between 378× for the lowest dilution (removing 99.8% of the human DNA) to 164× (removing 99.6% of the human DNA) in the highest diluted condition (1/15,000 *Leishmania*:human). 

Secondly, we wanted to test if enrichment via MBD columns worked equally well on *L. donovani* amastigotes for (a) fundamental reasons, as the (indirect) second method to detect if there are any methylation differences between promastigotes and amastigotes, and (b) practical reasons, as it is the (intracellular) life stage encountered in clinical samples. Therefore, we also carried out this enrichment technique on three sets (three strains) of hamster-derived amastigotes and their promastigote controls. Similarly, as in the previous experiment, *Leishmania*-human DNA mixes were generated in a 1/1500 (*Leishmania:*human) ratio, after which enrichment was carried out with the NEBNext Microbiome DNA Enrichment Kit. The enrichment worked well for both life stages, the promastigote samples were, on average, 76.22 ± 14.28 times enriched, and the amastigote samples 61.68 ± 4.23 times (Table 3).

## 4. Discussion

With this work, we present the first comprehensive study addressing the status of DNA-methylation in *Leishmania*. 

We demonstrated that the *Leishmania* genome contains a C5-*DNMT* (*LdDNMT*) that contains all 10 conserved *DNMT* domains. We also showed the gene is expressed at the RNA level. As the C5-DNMT family is diverse, and several family members are known to have adopted (partially) distinct functions during the course of evolution, we were particularly interested in the position of this DNMT within the evolutionary tree of this family, as it could direct hypotheses about the function of this protein. We found that LdDNMT is, in fact, a DNMT6 and that all studied *Leishmania* species and, more generally, the majority of trypanosomatids have exactly one copy of this DNMT6 in their genome (exc. *T. cruzi*) [20]. Interestingly, all other (non-trypanosomatid) species studied so far have either multiple *DNMT6* copies and/or other *DNMT* subfamily members in their genomes [20,75]. Therefore, *trypanosomatids* might be a unique model species to further study the role of this elusive DNMT subfamily, as there can be no interaction with the effects of other DNMTs. 

The fact that our LdDNMT knock-out line (verified by sequencing) was viable shows that DNMT6 is not essential for the survival of the parasite, at least in promastigotes and in our experimental conditions. However, at the same time, one might hypothesize that DNMT6 does offer a selective advantage to the parasite. First of all, the sequence of DNMT core domains is extremely conserved across the tree of life, and this is no different from those that we encountered in *Leishmania*. Secondly, *Leishmania* is characterized by high genome plasticity and features extensive gene copy number differences between strains [81,82]. Therefore, one might speculate that the parasite would have lost the gene a long time ago if it did not provide any selective advantage. 

In addition, we aimed to characterize the DNA methylation patterns of the parasite’s genome. Therefore, we carried out the first multi-life stage whole-genome bisulfite sequencing experiment on *Leishmania* and trypanosomatids in general. We checked both the promastigote (both culture and amastigote life stage). Surprisingly, we did not find any evidence for DNA methylation in *L. donovani* even though we checked both for large, regional patterns (sensitive for low levels of methylation over longer distances), and site-specific analyses (sensitive for high levels of methylation at individual sites). This could either mean that there is indeed no DNA methylation in these species or that was below our detection threshold. Regarding this detection threshold, two factors should be considered. Firstly, bisulfite sequencing and analysis allows for the detection of specific sites that are consistently methylated across the genomes of a mix of cells. In our case, we looked for sites that are methylated in at least 80% or 40% of the cases. Thus, if *Leishmania* consistently methylates certain genomic positions, our pipeline would have uncovered this. However, if this methylation would be more random, or occurring in only a small subset of cells, we would not be able to distinguish this for random sequencing errors, and as such, we cannot exclude this possibility. Secondly, bisulfite sequencing typically suffers from poor genomic coverage due to the harsh BS treatment of the DNA [83]. In our *L. donovani* samples, we covered at least 30.14% of the CpG sites, 29.47% of the CHG sites, and 24.23% of the CHH sites (even though having more than 90× average coverage). However, as there are millions of CpG, CHG, and CHH sites in the genome, the chance is very small (0.75^n^, with n = number of methylated sites) that we would not have detected methylated sites, even if present in low numbers. 

In any case, it is hard to imagine that any of the typical eukaryotic DNA methylation systems, such as genomic imprinting, chromosome inactivation, gene expression regulation, and/or the repression of transposable elements, could be of significance with such low methylation levels. On the other hand, given its phylogenetic position, it is perfectly possible that DNMT6 has changed its biological activity and now carries out another function. Indeed, as we described above, a similar phenomenon was observed with DNMT2 that switched its substrate from DNA to tRNA during the course of evolution [16,17]. 

Correspondingly, we did not observe any detectable DNA methylation for *T. brucei*. These findings are in contrast to what has been reported before by Militello et al., who detected 0.01% of 5MC in the *T. brucei* genome [27]. Besides, the methylated (orthologous) loci described in this paper could not be confirmed in the current work. However, this is maybe not surprising as the same authors reported later that TbDNMT might, in fact, methylate RNA, as they identified methylated sites in several tRNAs [71]. This would indeed explain why we did not observe C5-DNA methylation in *T. brucei* with high resolution, whole-genome bisulfite sequencing, and further suggest that a similar substrate switch to tRNA has occurred for DNMT6, just like has occurred for DNMT2. Further functional characterization of DNMT6 is required to verify this hypothesis.

From an applied perspective, this study opens new avenues for the enrichment of trypanosomatid DNA from clinical samples, which often have an abundance of host DNA. Indeed, depletion of methylated DNA could be included as a pre-enrichment step for existing enrichment approaches. For example, our group has recently obtained excellent sequencing results of clinical samples using SureSelect (97% of the samples for diagnostic SNPs, 83% for genome-wide information for sequenced samples), but was not able to sequence samples below 0.006% of *Leishmania* DNA content [73]. Perhaps the removal of methylated DNA could further enhance the sensitivity of this method. In the case of *Leishmania,* the technique could be useful for both enrichments from mammalian hosts and the insect vector, as it has been recently shown that the phlebotomine vector also carries Me^5^C in its genome [84]. The depletion of methylated DNA as a pre-enrichment step before whole-genome sequencing has also been successfully used before for the parasite *Plasmodium falciparum* (malaria) and shown to generate unbiased sequencing reads [85]. 

In conclusion, we demonstrated that the *Leishmania* genome encodes for a DNMT6, but DNA methylation is either absent or present in such a low proportion that it is unlikely to have a major functional role. Instead, we suggest that more investigation at the RNA level is required to address the function of DNMT6 in *Leishmania*. The absence of DNA methylation provides a new working tool for the enrichment of *Leishmania* DNA in clinical samples, thus facilitating future parasitological studies. 

## Figures and Tables

**Figure 1 microorganisms-08-01252-f001:**
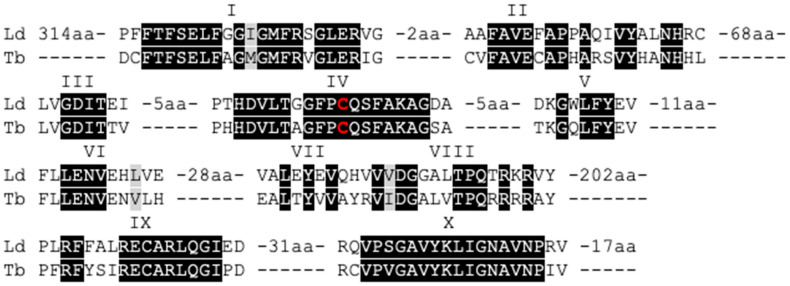
Protein alignment of LdDNMT (LdBPK_251230) and TbDNMT generated with T-coffee, picturing the similarities between the 10 homologous domains of C5 DNA methyltransferases. Black highlights homology, and the red character displays the position of the catalytic cysteine residue.

**Figure 2 microorganisms-08-01252-f002:**
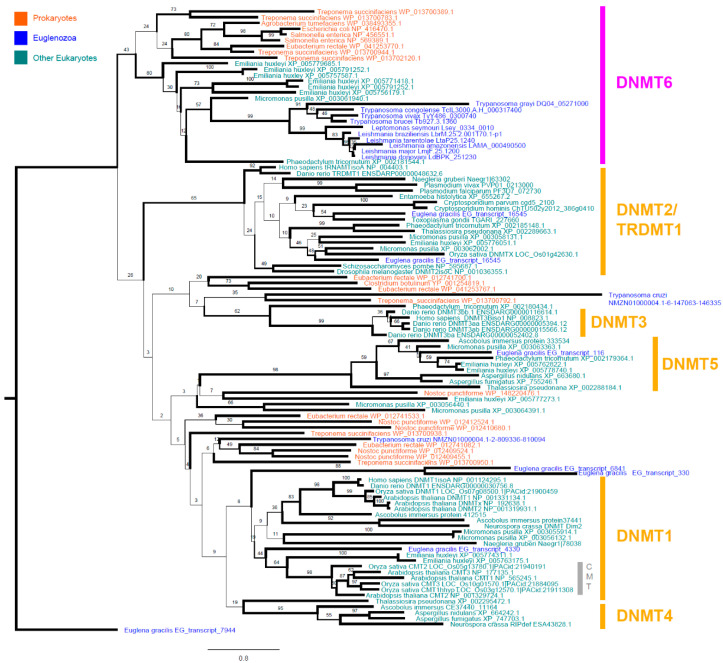
RAxML maximum likelihood tree showing the position of trypanosomatid DNMT (DNMT 6) within the DNMT family. Displayed branch bootstrap values are based on 1000 bootstraps. Line thickness is scaled for these bootstrap values.

**Figure 3 microorganisms-08-01252-f003:**
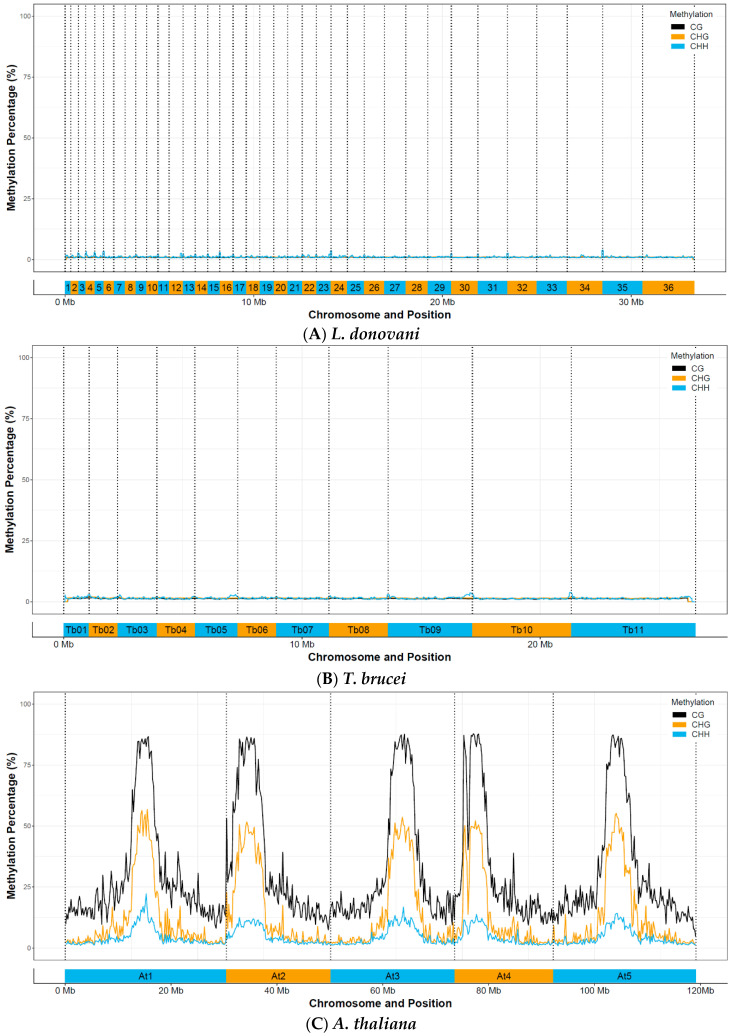
CpG, CHG, and CHH genome-wide methylation patterns in (**A**) *Leishmania donovani* BPK282 P3 promastigotes (36 chromosomes)*,* (**B**) *Trypanosoma brucei brucei* TREU927 (11 chromosomes), and (**C**) *Arabidopsis thaliana* Col-0 (5 chromosomes). Data was binned over 10,000 positions to remove local noise and variation.

**Figure 4 microorganisms-08-01252-f004:**
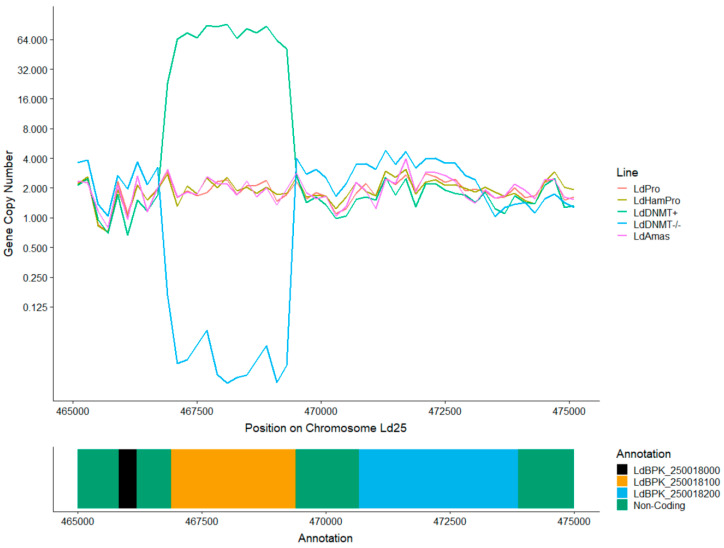
DNA/gene copy number based on genomic sequencing depth on chromosome 25 position 465,000–475,000. Both the LdDNMT knock-out (LdDNMT-/-) and LdDNMT overexpressor lines (LdDNMT+) were successful with, respectively, 0 and 64 copies of the gene. The plot shows also that the neighboring genes LdBPK_251220 and LdBPK_251240 were unaffected and had the standard disomic pattern.

**Figure 5 microorganisms-08-01252-f005:**
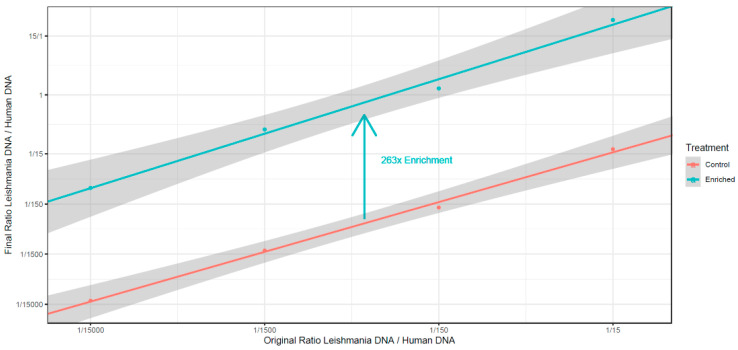
Enrichment (X) of *Leishmania* DNA in artificial mixtures of *Leishmania* promastigote DNA and human DNA, with the mixtures ranging from 1:15 to 1:15,000 *Leishmania:*human DNA. Enrichments were carried out with the NEBNext Microbiome DNA Enrichment Kit (NEB), and the unmethylated *Leishmania* DNA was enriched on average 263 times.

**Table 1 microorganisms-08-01252-t001:** CpG, CHG, and CHH methylation percentages in different *Leishmania donovani* lines (Ld), *Trypanosoma brucei,* and *Arabidopsis thaliana* (positive control).

	CpG (%)	CHG (%)	CHH (%)
**LdPro**	0.0003	0.0005	0.0126
**LdAmas**	0.0001	0.0003	0.0073
**LdHamPro**	0.0002	0.0005	0.0113
**LdDNMT+**	0.0013	0.0026	0.0627
**LdDNMT-/-**	0.0002	0.0006	0.0079
**Tbrucei**	0.0001	0.0006	0.0040
**Athaliana**	21.0473	4.0401	0.3141

**Table 2 microorganisms-08-01252-t002:** qPCR estimation of LdBPK_251230 expression level (copy number) of Ldo-Pro and Ldo-DNMToverex.

	Ldo-Pro	LdDNMT+
RNA	1.53 ± 0.2	3.78 ± 0.3

**Table 3 microorganisms-08-01252-t003:** Enrichment (X) of *Leishmania* DNA in artificial mixtures of *Leishmania* and human DNA for promastigotes and amastigotes of 3 clinical isolates (BPK026, BPK275, and BPK282). Enrichments were carried out with the NEBNext Microbiome DNA Enrichment Kit (NEB).

	BPK026	BPK275	BPK282	Average Enrichment (X)	St.Dev
Promastigotes	79.85	88.32	60.47	76.22	14.28
Amastigotes	64.83	56.87	63.33	61.68	4.23

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
