# Peer review of "The Absence of C-5 DNA Methylation in Leishmania donovani Allows DNA Enrichment from Complex Samples"

_microorganisms, 2020, doi:10.3390/microorganisms8081252_

Round 1

Reviewer 1 Report

In this manuscript the authors investigate C5 DNA methylation in Leishmania. Using bisulfite sequencing they show that C5 DNA methylation of genomic DNA is undetectable in Leishmania and that the deletion or overexpression of the DNMT enzyme does not affect this. They utilise the difference in gDNA methylation between humans and Leishmania to enrich for Leishmania gDNA. Overall, this manuscript is well-written and the data supports the conclusions and I only have minor comments for the authors.

I am a bit confused by the analysis performed for figure 2. They are looking at the evolutionary profile of the DNMTs and show that the Leishmania/Trypanosoma DNMT are grouping together and they are close to the prokaryotic DNMT. However, based on a previous analysis they say that the Leishmania/Trypanosoma DNMTs are part of group 6 but their analysis does not have any group 4/5 sequences on it nor does it have any other potential group 6 sequences from diatoms that you would expect to group with the Leishmania/Trypanosoma sequences – it would be good to include those sequences to show definitively that the Leishmania/Trypanosoma DNMT is in group 6. What result does this evolutionary analysis provide over the one described in reference 20? If it is showing where the DNMTs from other diverse eukaryotes group that is useful but they should include the full range of DNMT sequences to get a clearer picture of how these DNMTs from diverse organism group.

Does the close evolutionary relationship between group 6 DNMTs and prokaryotic DNMTs come from horizontal gene transfer? Is there evidence for this?

What is the function of the prokaryotic DNMT? Does this give clues to the potential substrate of the Leishmania DNMT?

The authors show that in their DNMT+ strain there are multiple copies of the gene present and there is an increase in mRNA but without assessing the protein expression it is difficult to be sure that this line is in fact overexpressing the protein, which in the end is the point for such an experiment.

Did the deletion or overexpression of DNMT have any effect on the growth of the parasites?

In the results section it would be good to include the names of the genes that were used to assess the relative amounts of gDNA present from Leishmania and humans. Why did the authors use cysteine synthase? Is cysteine synthase a single copy gene? Is it known to be on chromosome prone to polyploidy?

Line 135 should be 5 x 105

Line 157 should be 5 x 107

Line 321 – I presume it’s the mRNA level?

Reviewer 2 Report

In this study, Cuypers et al analyzed DNA methylation using whole-genome bisulfite sequencing in Leishmania donovani in both life stages, promastigote and amastigote. Remarkably, no clear evidence of DNA methylation was observed, even after overexpressing a putative DNA methyltransferase present in the L. donovani genome. Moreover, deletion of the gene coding for this DNA methyltransferase was well-tolerated by the parasite, suggesting a non-essential role of this protein, at least during axenic growth. Finally, based on the low levels (if any) of DNA methylation in the Leishmania genome, the authors explored the use of Methyl-CpG-binding domain columns (commercially available) for parasite DNA enrichment from samples having a mixture of parasite and host DNA. The results support the use of this strategy for analyzing parasite DNA in clinical samples.

The manuscript is well-written, the research is original and the work is technically sound.

I have minor questions that may be considered by the authors when revising the manuscript.

Specific points.

  1. Page 3, line 134. It is indicated that the amastigotes were obtained as described in Ref. 29. Nevertheless, in ref. 29, it is stated that amastigotes were obtained as described in: Pescher P, Blisnick T, Bastin P, Späth GF. 2011. Quantitative proteome profiling informs on phenotypic traits that adapt Leishmania donovani for axenic and intracellular proliferation. Cell Microbiol 13:978 –991.

            This reviewer has not checked if the latter reference contains a detailed description of the method. Nevertheless, taking into account the relevance of the method for this study, authors may consider including a brief description of the method, paying attention to the processes of liver homogenization, amount of sample loaded, how the Percol gradient was made, and centrifugation conditions.

  1. Page 4, line 173 (and other places). This reviewer could not find the entry LdBPK_250018100.1 either at TriTryp or NCBI databases. Could the authors specify (perhaps by a web link) where the information regarding this gene can be obtained? Otherwise, the authors may consider using the code LdBPK_251230 (page 4, line 154), which is searchable in the mentioned databases.

  1. Page 4, line 173. It would be more appropriate to indicate upstream and downstream sequences instead of 5' and 3'-UTR, at least that the authors know that both UTRs are longer than 300 bp.

  1. Page 5, lines 204-205. Please double-check the concentration values for Leishmania and human DNA in the artificial mix (1/1500).

  1. Page 6, line 230. Please, check the correctness of epigraph 3.2.

  1. References 12, 29, 46 and 69 seem to be incomplete.

Reviewer 3 Report

DNA methylation enzymes are nowadays studied as they are active in the restriction modification systems, with a huge role in the defence of organisms against phages and viruses. The manuscript entitled „The absence of C-5 DNA methylation in Leishmania donovani allows DNA enrichment from complex samples’bu
Cuypers B,et al is describing the status of C-5 DNA methylation in Trypanosomatids with focus on Leishmania, with a comprehensive study of genomic methylation in Leishmania across different parasite life stages, by using the high-resolution whole genome bisulfite sequencing.
C-5 cytosine-specific
Firstly the authots demonstrated the presence of C5 DNA MTases DNA methylase by searching of the parasite’s genome by using the LdBPKv2 reference genome and by searching the predicted protein sequences of the assembly using the hidden-markov-model (hmm) signature of the C5 DNA MTase protein family obtained from PFAM (PF00145) and obtaining a single hit: the protein LdBPK_250018100.1 (LdDNMT).
Then 131 putative family members in the genomes of 24 species were identified, including 4 Prokaryoticand 20 Eukaryotic species. The phylogenetic tree exhibited known DNMT subgroups, including DNMT1, DNMT2, DNMT3, DRM (Domain rearranged methyltransferase), DIM and 2 groups of Prokaryotic DNMTs.
Afterthat, the functional role of C5 DNA methylation in Leishmania donovani was elucidated by assessing the locations and degree of CpG, CHG and CHH methylation across the entire Leishmania genome and within the two parasite life stages: amastigotes (intracellular mammalian life stage) and promastigotes (extracellular, insect life stage).
After cheking the methylation patterns across the genome, no methylated regions in Leishmania donovani were identified. Then the authors proceed to search for the fully methylated sites by using BS-Seeker2 and filtering the results with the automated Python3 workflow, and the data was evaluated even manually. The sequenced of L. donovani DNMT knock-out (LdDNMT-/-) line and that of DNMT overexpressor (LdDNMT+) were also performed to clarify the importance of LdDNMT for C-5 DNA-methylation pattern. Absence of C5 DNA Methylation as a Leishmania vs host DNA enrichment strategy was evaluated.
The data presented are strong, and convincingly demonstrated the presence of 10 conserved DNMT domains in Leishmania genome and that the gene is expressed at the RNA level. The manuscript is concise and the appropriate analyses and use of controls are performed.
This is a well performed study that I consider that is important and represent a reliable strategy to characterise the DNA-methylation patterns of a parasite’s genome, with influence on future applications for preparing the clinical samples.

Minor revision
The authors need to address the below comments to strengthen quality of the manuscript:
Please rewrite the phrase from lines 347-348: “To test this if these kits can be used for Leishmania, we first generated artificially mixed samples using different ratios of L. donovani promastigote DNA with human DNA.”
